# A Human Capability Perspective on the Progression of Low-SES Students to Higher Education in Ireland and the UK

Cliona Hannon

Trinity Access Programmes, Trinity College Dublin, University of Dublin, D02 PN40 Dublin, Ireland; clionahannon2014@gmail.com

**Abstract:** This article focuses on targeted programs for low-SES students in two selective universities: Trinity College Dublin, the University of Dublin, Ireland (Trinity Access Programmes/TAP) and the University of Oxford, UK (Lady Margaret Hall Foundation Year/LMH FY). The programs were collaborative developments, as examples of the potential of learning and adaptation across geographical contexts. It poses two questions: (a) How did the admissions processes in both universities change to target low-SES students? (b) How do social and academic support services for low-SES students, provided by two universities, contribute to the development of student capabilities? The article draws on the capability approach as the evaluative lens used to explore the two programs. Findings indicate (a) innovative approaches to socio-economic assessment in both programs, resulting in effective targeting of low-SES students, (b) the scaling of the programs beyond their initial remit and (c) the emergence of specific student capabilities through their engagement in the programs.

**Keywords:** capability; education; inequality; justice; socio-economic status

## 1. Introduction

Higher education participation rates have reached a record high across the Organisation for Economic Cooperation and Development (OECD) countries. Young adults with degree-level qualifications constituted 48% of 25–34-year-olds in 2021 and reached rates of 69% in Korea and 66% in Canada. Despite this expansion across the OECD, a low-SES young person is less than one-half as likely to be in higher education compared with the proportion of low-SES families in the population. This compares with a high-SES young person, who is almost twice as likely to be in higher education [1]. The disparity in academic achievement between students from low-SES and high-SES backgrounds—the 'socio-economic achievement gap'—is documented across a wide range of countries, and there is evidence it has increased in the last 50 years [2].

The economic returns of a higher education remain high. OECD evidence shows that, for young people, the higher their educational attainment at the start of the last economic crisis, the more likely they were to be employed throughout the Great Recession [3]. In Ireland, graduates hold almost a half of all jobs, although they comprise only one-third of the working age population, and their employment rate is 80%, against a 61% rate for the population at large [4]. Higher education confers other social and cultural returns, including better health, longer life expectancy and greater life satisfaction. Graduates are also more likely to participate in society through voting and volunteering [5].

While progression to higher education is the outcome of complex, interwoven factors both within and outside of school, many universities across the OECD have established targeted programs to tackled inequalities in higher education progression. This article is a comparative case study of the impact of two such programs: Trinity College Dublin (Trinity Access Progrmames/TAP), Ireland and The University of Oxford (Lady Margaret Hall Foundation Year/LMH FY), UK. The programs were collaborative developments, as examples of the potential of learning and adaptation across geographical contexts.

This article draws on the capability approach as the evaluative lens used to explore the qualitative outcomes of both programs. It poses two questions: (a) How did the admissions processes in both universities change to target low-SES students? (b) How do social and academic support services for low-SES students, provided by two universities, contribute to the development of student capabilities?

The article begins with an overview of 'widening participation' programs to improve access to higher education by low-SES groups in the UK and Ireland. It explores how the admissions systems within both universities were adapted to target low-SES students. It explains the materials and methods used in evaluating the two programs. It continues with an explanation of the emergent capabilities in low-SES students in both programs. Finally, it considers the contribution of the capability approach as an evaluative lens and the benefits of cross-national collaboration in tackling low-SES student higher education progression.

## 2. Widening Participation in Ireland and the UK

This section explains the access to higher education (HE) landscape in Ireland and the UK. Higher education access policy in both countries was developed in the late 1990s under the term 'widening participation' (WP). The approach to WP in both countries shares similarities. 'Targeted programs' between schools and universities were funded by government from the 1990s onwards. In both cases, regulatory frameworks have emerged to monitor and incentivise HEIs to further diversify their admissions intake. Unlike the UK, Ireland has remained a comparatively low-cost HE system, and maintenance grants are still available to low-SES students. While Ireland has a much smaller HE system, there has been much greater collaboration across the Irish HE system in developing shared admissions systems to target low-SES students.

### 2.1. Access to Higher Education in Ireland

In Ireland, 93% of students attend a state secondary school and 7% attend a private (fee-paying) or independent school. Ireland has recently launched its fourth National Access Plan. Secondary-school completion rates have risen to 90% and 64% of young adults progress to higher education [6]; however, a number of surveys show that there are significant, and persistent, disparities in social class participation in higher education [6–12]. Twenty-seven per cent of low-SES students progressed to higher education in 2017 [13]. In 2020, there were 4.9 students from disadvantaged areas to every 10 students from affluent areas [6].

Higher education institutions (HEIs) focus on widening the participation rates of low-SES students through a combination of educational outreach programmes and diversified admissions routes. The standard outreach model provides a range of educational opportunities, including visits to the HEI's campus, learning supports, foundation courses to help bridge the gap between school and HE, and contextual admission routes to access "targeted" places in HE institutions.

Ireland embarked on a policy of funding "targeted" widening participation programmes between schools, community groups and HEIs in the late 1990s. These programmes focused on sustained outreach to schools and communities, alongside the development of contextualised admissions systems, which allowed for admission of low-SES students at lower tariffs than those from more advantaged backgrounds. HEIs agree targets for students from low-SES backgrounds as part of their "performance based compact" with the Higher Education Authority (HEA). The compact is an agreement across a range of institutional metrics, and it forms the basis of discussions between each HEI and the HEA to determine state investment levels. This is similar to the role played by the Office for Students (OfS) in England in incentivising institutions to diversify their intake.

### 2.2. Access to Higher Education in the UK

Similar to Ireland, higher education institutions in the UK developed localised responses to improve their socio-economic diversity, and research demonstrates these inter-

ventions were effective [14]. The current progression rate to HE in the UK is 37% [5]. The cost of HE to students was relatively low-cost, similar to Ireland, until 2010, when the then coalition government precipitated substantial changes to the HE funding model, raising the tuition fee limit to GBP 9000, while larger government-funded loans were provided to students taking a first degree. Subsequent policy changes mean that HE maintenance grants are no longer available to students, regardless of their SES. Institutions charging the current GBP 9250 tuition fee tariff are required to produce an access agreement outlining how they will use their additional revenue to support access measures. These are normally set out in an agreement between the Office for Students (OfS) and each institution [15]. This means the UK now has one of the most expensive tuition fee systems in the world for a primary degree and young people can anticipate leaving university with a debt burden of at least GBP 50,000.

Not all universities are created equal, and the policy discourse around access to HE in the UK has shifted in the last decade to a closer consideration of "fair access" for low-SES students to selective higher education institutions. Prestige differences emerged between the "Old" (pre-1992) and "New" (post-1992) universities [16–18]. Graduates of more selective universities are more likely to secure professional and managerial jobs and to earn higher salaries [19,20]. Data show that students from low-SES backgrounds who applied to university in 2020 were four times less likely to go to a highly selective university than students from the most advantaged backgrounds [5].

In 2018, the government established the Office for Students (OfS) as the regulator of higher education in England. It took on responsibility for promoting "fair access" to higher education and ensuring students from all backgrounds are supported to progress and graduate. The OfS introduced Access and Participation Plans (APPs) as a condition for HEIs who wished to charge higher tuition fees. APPs required HEIs to demonstrate the range of on-campus and outreach activities they take to 'widen participation'. HEIs are also encouraged to make use of "contextual data" in their admissions decisions. This process involves considering an individual's socio-economic background and their school performance data, among other things, rather than relying solely on their results in exams and coursework [21]. Contextualised admission varies across countries. Mountford-Zimdars and Sabbagh [22] explain the holistic admissions approach taken in the USA, where each student is considered individually. In the UK and Ireland, on the other hand, the admissions system draws on "indicators" of disadvantage. Four types of indicators are generally used: individual, postal code-level, school-level and participation in widening participation programs [18].

This section has explained the context in Ireland and the UK for low-SES student progression to higher education. The following section outlines the materials and methods used to explore two questions posed in relation to two educational interventions: (a) the admissions adjustments required in both countries to diversify low-SES student intake and (b) the developing capability set of students participating in a community-based educational outreach programme (TA21) and a university-based academic programme (LMH FY).

## 3. Materials and Methods

This section explains the use of a comparative case study to explore two programs: a community-based programme, Trinity Access 21 (TA21), Trinity College Dublin and a university-based programme, Lady Margaret Hall Foundation Year (LMH FY), the University of Oxford. A case study is "an in-depth examination, over time, of a single case", whereas comparative case studies "cover two of more cases in a way that produces more generalisable knowledge about how and why particular programmes or policies work or fail to work" [23]. A comparative case study is useful to understand how context influences the implementation of an intervention. This section also provides an overview of the capability approach as the exploratory lens applied to the qualitative outcomes of both programs.

This comparative case study (CCS) poses two questions: (1) How did the admissions processes in both universities change to target low-SES students? (2) How do social and academic support services for low-SES students, provided by two universities, contribute to the development of student capabilities? The CCS is undertaken in the context of a collaboration between Trinity College Dublin and LMH Oxford University (2016–2020) in the development of a foundation year and an adapted admissions process. TA21 developed from an experienced team (Trinity Access Programmes/TAP), which had built a range of community- and university-based programs and admissions processes over a 20-year period. The LMH FY developed in partnership with TAP but from a team that had limited experience in developing such programs. Despite this, the LMH community had a strong commitment to improving diversity and leading admissions change in the University of Oxford.

The TA21 (community-based) developed in response to team experience and the research evidence base, which indicated a range of challenges for low-SES students in second-level education. Smyth and McCoy note that schools in areas of low-SES have a higher proportion of "newcomer" (immigrant) students, students with disabilities and traveller students than schools in more affluent areas and have a higher incidence of serious literacy and numeracy problems, emotional and behavioural problems, absenteeism, lower student motivation, problematic student–teacher relationships and less parental involvement [24].

The LMH Foundation Year (university-based) arose in the context of persistent inequalities in accessing selective universities in the UK and the research evidence base, which indicated that barriers to greater equality in higher education included the prior academic attainment of students, inadequate advice and support during school and financial concerns about the cost of higher education.

The CCS draws on the recent research and evidence base for both programs in the form of published peer-reviewed articles and internal university publications and reports. It uses a capability approach lens to explore the available qualitative findings on the impact of both programs on participating students. It also draws on the relevant literature and policy context within both domains. Table 1 below describes the key features of each programme:

**Table 1.** Targeted programs in Ireland and the UK to improve access to HE for low-SES students.

|  | Community-Based within Schools: TA21 | University-Based within a College: LMH FY |
|---|---|---|
| Programme lead | Trinity Access Programmes (TAP), Trinity College Dublin (TCD). | Lady Margaret Hall, Oxford University, in partnership with Trinity Access Programmes (TAP), TCD. |
| Aims | To provide every student in a "Leader" school with multiple opportunities to receive (a) educational advice and guidance, (b) relatable educational role models, and (c) opportunities to identify and lead change in the classroom and in the wider school community. To undertake an action research study on the programme and to use the outcomes for (a) school improvement and planning and (b) national policy change. | To provide low-SES students aiming to progress to Oxford University with a personal, social and academic foundation (the "foundation year" or FY) to enable them to flourish in higher education. To reform the low-SES admissions system in LMH and Oxford University using the student selection experience from the FY. To scale the FY beyond from LMH to across Oxford University. |
| Description | TA21 provides a "whole school approach" to university–school widening participation interventions, involving every student from year one of second-level schooling (and throughout) in three "core practices": (a) Pathways to College, (b) Mentoring and (c) Leadership in Learning. Schools agree a framework for the three "core practices" and the research across all year groups and participate in a programme network that includes teacher continuous professional development (CPD). | LMH developed a year-long foundation year that includes 16 subjects, a blend of academic content and preparation for undergraduate studies. FY students are full members of the college community. FY students may then matriculate to Oxford University provided they meet set academic standards on the FY. |

**Table 1.** *Cont.*

|  | **Community-Based within Schools: TA21** | **University-Based within a College: LMH FY** |
|---|---|---|
| Target group | Second-level students ages 12–18. | Second-level students (age 18–19) who have recently completed their A-levels and who have strong academic ability but are unlikely to be admitted directly to Oxford University, as their A-level grade achievements have been affected by their socio-economic context. |
| Geographic focus | Mainly concentrated in Dublin but includes schools across Ireland. A total of 20 "Leader" schools and 40 "Network" schools participate each year. | Low-SES students ages 18–19 drawn from across the UK. Approximately 10 students are selected for the programme each year. |
| Sampling for qualitative studies | Purposive sampling: 35 students from 4 different school types (all low-SES), male and female and low-SES family backgrounds. | Purposive sampling: all students on the LMH FY. |
| Number of students involved in qualitative studies | 35 | 10 |
| Evidence base | References [25–33] | References [34–40]. |
| Emergent student capability set after programme | Autonomy, hope, identity, social relations and social networks, practical reason and knowledge. | Identity, social relations and social networks, practical reason and knowledge. |

*3.1. Overview of Comparative Case Study Programme A: Trinity Access 21*

This section explains the TA21 programme, which is the main educational outreach programme within Trinity Access Programmes (TAP), Trinity College Dublin. The TA21 programme evolved from a 20-year evidence base in practice and policy to improve low-SES progression to higher education. It aimed to address two problems: first, that in addition to the educational barriers caused by poverty, there was limited educational guidance, subject choice and educational role models within some schools. Second, the post-primary curriculum is considered to be inflexible and focused on the terminal exams [25–27].

The overarching goal of TA21 is to address these two problems through the delivery of a suite of programmes for students and the provision of professional development for teachers, both of which are embedded in an iterative cycle of data collection and analysis, feedback to schools and consultation with stakeholders [25].

There are 20 "leader" schools in the project, all of which are post-primary schools located in low-SES communities. In addition, there are 40 linked "network" schools associated with the project. These are spread over a wide geographic area and receive less direct support from the team. The "leader" and "network" schools approach allows the project to scale both deep and wide and to use learning from the scaling deep in scaling more broadly within the "network" schools.

The TA21 project is underpinned by a "widening capability" [29] model of widening participation and aims to shift from a focus solely on student progression to one that includes student potential and capability. TA21 draws on a US intervention, CFES Brilliant Pathways, which has three components derived from the theory of Academic Capital Formation [29] that aim to increase students' understanding of college application and support services (Pathways to College), provide individual mentoring of students (Mentoring) and develop students' leadership skills (Leadership through Service). A fourth aspect—twenty-first century (21C) teaching and learning—was added to the original CFES model with the goal of supporting teachers to transform their pedagogical approaches to help develop students' key 21C skills. This final component was combined with Leadership through Service, as there was significant overlap between many of the skills that were developed through participation in both practices. The new core practice has been termed Leadership in Learning [26].

This subsection has explained the TA21 programme. The following subsection explains the LMH FY programme.

*3.2. Overview of Comparative Case Study Programme B: Lady Margaret Hall (LMH) Foundation Year, the University of Oxford*

This subsection explains the University of Oxford admissions context within the wider UK policy landscape and as the backdrop for the development of the LMH FY. In the UK, approximately 93% of A-Level students attend state schools, while the remaining 7% attend independent (fee-paying) schools. Research shows that 30% of state schools have at most one or two students progressing to the prestigious Russell Group universities in the UK, with just 40 schools and colleges providing a quarter of all Oxbridge (Oxbridge is a combined term used in the UK to describe the Universities of Oxford and Cambridge) entrants [34]. The University of Oxford admissions report (2022) shows significant geographical bias to South East England, and the majority of admissions are to students with professional parents [34,35].

Oxford and Cambridge universities have significantly increased their WP activity over the last decade, but this has not had the effect of diversifying its socio-economic intake. Admissions to Oxford are complex, as it is a collegiate university comprising 38 colleges and 6 Permanent Private Halls (PPHs) of religious foundation. Every student must be a member of a college or a PPH. Each college is an autonomous, self-governing institution with responsibility for its own admissions. Admissions are carried out in accordance with the Oxford University Common Framework of 2006 [34].

Therefore, admissions decisions are taken at the level of individual colleges and PPHs, each one of which has significant latitude in planning for WP. Some colleges provide for greater flexibility over minimum entry requirements and academic attainment for students from low-SES backgrounds [34].

Lady Margaret Hall, a college of the University of Oxford, partnered with Trinity College Dublin to adapt a year-long intensive academic preparation course (the Foundation Year or FY) to that context. The goal of this project was to increase socio-economic diversity in the University of Oxford and to prove that students from low-SES backgrounds could reach their full academic potential within a well-supported environment. A secondary goal was to scale the pilot project beyond LMH to other colleges within the University of Oxford collegiate structure.

Universities in Ireland and England have developed FY programmes as a way of supporting students to transition into university and to supplement "academic attainment gaps" at second level. These are intended for students without the formal entry qualifications for their chosen degree. They are designed to prepare entrants for degree-level study [35]. FY programmes recognise that the challenges facing low-SES students in HE are complex and focus on the importance of supporting the development of peer relationships, academic skills and a sense of belonging in the university.

In 2016, a group from Lady Margaret Hall (LMH), a college of the University of Oxford, visited Trinity College Dublin (TCD), the most selective higher education institution in Ireland, to explore the potential of adapting the Trinity Access Programme (TAP) Foundation Course to the Oxford context. The TAP Foundation Course (TAP FC) for Higher Education was established in 1997 in Trinity College Dublin (TCD) as a year-long, intensive academic preparation course. It was developed to provide an alternative matriculation into the university for low-SES students who had not reached their academic potential for socio-economic reasons. Progression to the university depended on successfully completing the course and meeting specific academic requirements. The TAP FC provides social and financial supports for these students throughout their time in the University. At the time of LMH's visit to Ireland, TAP had developed a 17-year-long evidence base, with over 90% of TAP FC students successfully progressing to the university and 89% of those who had progressed successfully completing a degree. The positive story of the TAP FC, which had been captured from the outset through research and evaluation, therefore presented a strong model from which to develop a Foundation Year in Lady Margaret Hall (LMH) [35].

The governing body of Lady Margaret Hall articulated a strong desire to widen their admissions to socio-economic groups and areas of the country from which they did not

have robust representation. LMH had its historical foundations in taking radical action on admissions, as it was the first women's college within Oxford University; however, developing this kind of alternative entry route was a challenge to the university. It had been agreed by LMH's Governing Body that Foundation Year students would be admitted to the programme on lower academic attainment that those being admitted to first year in the university. The rationale for this was that the Foundation Year would target students from backgrounds where they had not had the opportunity to reach their full academic potential. This approach was considered by some within the university to represent a deviation from the common framework on admissions (Robson et al., 2017). Despite internal challenges in Oxford, in 2016 TCD and LMH established a partnership to adapt the TAP Foundation Course to the Oxford University context as part of a four-year pilot. In the 2016–2017 academic year, within an overall intake of 670 students, 10 Foundation Year students were registered for the inaugural LMH Foundation Year (LMH FY) [35].

The LMH FY is a year-long academic programme consisting of 16 subjects, including a range of sciences and humanities, and students also take a subject-specific course. The students have group teaching that includes the core modules of academic writing and preparation for undergraduate study. Students sit an interview as well as an examination to be considered eligible for the degree course they wish to progress into following the FY. Lectures by guest speakers and educational visits occur throughout the FY and these sessions are shared with undergraduate students. Students on the LMH FY who wish to progress to the University of Oxford for undergraduate study have to go through the standard admissions process for undergraduates, including admissions tests (subject-dependent) and interviews. If they are made an offer, it is conditional on successful completion of the LMH FY to an agreed academic standard. This section has described the key features of the TA21 and LMH FY programs within the CCS. The next section explains the capability approach, which was used as an exploratory lens with the qualitative data from both programs.

### 3.3. Using the "Capability Approach" to Explore the Qualitative Data

Drawing on the qualitative data from both of the CCS programs, a capability approach is used as an exploratory lens. Robeyns [41] argues that regardless of whether we focus on structure or agency, individual well-being should be the "ultimate unit of moral concern". The capability approach is a theoretical framework with two core-normative claims. First, the claim that freedom to achieve well-being is of primary moral importance and, second, that this freedom is to be understood in terms of individual capabilities, that is, their real opportunity is to do and be what they have reason to value [42]. An individual's capability represents their freedom or real opportunity set [43].

In recent times, the capability approach has been used to explore policy and practice in education, as an alternative framework to human capital theory, through which the process, purpose and impact of education can be evaluated [44–46]. Developed by Amartya Sen [47–49] and subsequently elaborated by Martha Nussbaum [50–52], the key idea of the capability approach is that social arrangements should aim to expand people's capabilities, which is their freedom to promote or achieve functionings that are important to them.

A key concept in the approach is people's "functionings": these are the "beings" or "doings" that are important to them, such as being able to access adequate food, accommodation or having time to read. The capability to achieve a functioning depends on a range of personal and social factors that vary across geographies and contexts. Focusing on the individual's *capability to achieve a functioning* rather than the function alone recognises the different circumstances of people and their varying preferences [53]. The capability approach has four central concepts [54–56]. These are:

1.  Capabilities: the freedom an individual has to enjoy valuable functionings. Capabilities are the alternative combinations of functionings that are feasible for a person to achieve.
2.  Functionings: beings or doings that an individual values (achieved outcomes). Examples of functionings can vary from being healthy or having a good job, to more

complex states, such as being happy or having self-respect. A functioning is the active realisation of capabilities.

3.  Agency: the ability of an individual to realise the goals that they value.
4.  Well-being: the appropriate focus for assessment of how well an individual is doing, defined by the links between material, mental and social well-being.

Individuals differ in their capacity to convert capabilities into functionings and conversion factors, such as structural or social arrangements, influence their exercise of agency.

*3.4. Human Capital and Human Capability*

Human capital theory is relevant to higher education access as it posits that by enabling more people to access higher education and to achieve post-secondary qualifications, we will generate a greater economic contribution and therefore create more opportunities for all.

Melanie Walker [57] has written extensively on the limitations of a human capital approach to education in the South African and UK contexts. She argues that it assumes labour markets are efficient at placing people in work suitable to their skills and that opportunities are shared equally. Walker [57] remarks that developing widening access to higher education is primarily useful in building human capital and is a persuasive and verifiable, market-aligned model, but it offers an impoverished model for education as it does not prioritise well-being, human agency or the transformative potential of education. She concludes that the human capital model is not sufficiently expansive to address social justice, given that what is really at stake in the model is economic growth.

Taking a capability approach, the dimension for measuring equality would be each person's capabilities. Walker [58] advocates that we ask the question "What is each person however able to be and to do?" rather than "What resources do they have?" Policy, Sen argues, should not just aim to increase income and educational qualifications but to increase access to the resources that enable these freedoms [59].

From this perspective, for instance, it may be that some students within schools with low progression rates do not choose higher education because they are "low achievers" or have "low aspirations" but instead they have "different aspirations". The freedom to choose higher education must include the freedom to reject it [60]; however, it is important to consider the context within which decisions are made and whether or not adequate information, guidance and support have been available. The capability approach invites a range of more searching questions about equality than just a focus on desire satisfaction. It is a challenge not just to evaluate resources and inputs, but also to consider whether learners are able to convert resources into capabilities and thereafter potentially into functionings. This section has provided an overview of both of the programs within the CCS and the materials and methods used to compare their impact in Ireland and the UK. It also explained the strengths and purpose of applying a "capability approach" lens to qualitative findings in both programs. The following section elaborates on the outcomes of this capability approach exploration and compares other key outcomes from both programs.

## 4. Results and Discussion

This section explains the outcomes from the comparative case study (CCS) which considers two programs: Programme A—TA21, a whole-school, educational outreach programme and Programme B—LMH FY, the collaborative development of a foundation year for low-SES students, between Trinity College and LMH in the University of Oxford (2016–20). It also compares the development of contextualized admissions processes in Ireland and the UK. It discusses how these were framed by and within specific policy contexts and how TA21 operated as a community-based programme focused on change at a school level, whereas LMH FY was a university-based programme, focused on change in higher education.

This CCS poses two questions:

(1)  How did the admissions processes in both universities change to target low-SES students?

(2)    How do social and academic support services for low-SES students, provided by two universities, contribute to the development of student capabilities?

Table 2 below identifies the key features of comparison across the two programs, and these are then explained.

**Table 2.** Comparing change between a community-based and a university-based education intervention to improve low-SES higher education access.

|  | Community-Based Change TA21 | University-Based Change LMH FY |
|---|---|---|
| 1 | Socio-economic indicators | Socio-economic indicators |
| 2 | Focused on community- and system-level change | Focused on diversifying SEG profile of university |
| 3 | Evolution of practice based on local evidence base | Adaptation of practice from another country or institution, customised to local need |
| 4 | University as change agent in partnership with schools, communities and students | University responding to external policy requirements and some internal pressure to diversify |
| 5 | Using capability lens to explore lived experience of students, drawing on research to re-frame practice with schools as partners | Using capability lens to explore lived experience of students, drawing on research to persuade faculty and administration of the viability of the FY as an admission route |

*4.1. A Comparison of Socio-Economic Indicators Used in Admissions Processes in Both Universities to Target Low-SES Students for HE Admissions*

While national policy had incentivised the development of "targeted programs" for low-SES students in Ireland and the UK, in the early 2000s, there was no agreed definition of "educational disadvantage" or the low-SES students who were to be targeted, and neither was there a system within the universities for doing so. In both programs, there was a need to define the socio-economic indicators that would serve as proxies for educational disadvantage. Trinity College Dublin had developed a set of national policy-based indicators for the TAP FC. The Higher Education Access Route (HEAR) emerged from this evidence base, and it was built and tested in close collaboration with two other universities: University College Dublin and Maynooth University [37]. Both universities and the Dublin Institute of Technology collaborated to develop a nationwide systems-based scheme. It was agreed across the sector that no single indicator would be sufficiently robust to qualify a student as eligible to compete for the targeted places in HEIs. There were both individual- and group-level indicators and they combined financial and socio-cultural proxies. Six indicators were ultimately agreed upon across the HE sector, and students had to meet at least three indicators to compete for a targeted HE place. HEAR was scaled from a local, manual process to a university-wide system in 2012. It is mainly through HEAR that each university recruits low-SES students to meet their SES targets. The six socio-economic indicators used in the scheme combined individual (e.g., family income) and group-level (e.g., geo-code) information linked to "disadvantage", which together functioned as a "proxy definition" of educational disadvantage.

This was an important development in moving contextualized admissions from a local, manual process with considerable subjective input on a university-by-university basis, to a systems-based, nationwide online application system that combined individual- and group-level data. The rationale for including both individual- and group-level indicators is that it enables more effective targeting of HE admissions processes at those low-SES students who need it. For instance, not all individuals living within a low-participation postal code is from a low-SES background, and there are low-SES students within high-attainment schools, and high-attainment postal codes. Combining a range of indicators enables a more

effective targeting of the students who have had a complex and disadvantaging experience of education. These were represented in Table 3 below.

**Table 3.** The Higher Education Access Route socio-economic indicators for low-SES students [27].

| Financial Indicators | Socio-Cultural Indicators |
| --- | --- |
| Income (students must meet a set threshold) | Socio-economic group |
| Medical card | School type |
| Means-tested social welfare payment | Postal code |
| | Also considered: In the care of the State |

This system has now been in use across the Irish HE system since 2012, and it is integrated within the shared application infrastructure used by all HEIs. Students apply to any HEI and provide documentation which is uploaded to the system and reviewed by a collaborative team nationwide. Students must meet certain academic thresholds for each course and institution in addition to the socio-economic indicators.

This learning was used by LMH in Oxford, where there was no agreed set of common indicators for selecting low-SES students across the university and there was an anecdotal view that the students who were being selected as "low SES" did not meet that definition. This was because the indicators used to assess socio-economic status leaned towards group-level indicators, such as POLAR or ACORN (school and postal-code progression), rather than integrating a focus on the SES of the applicant. This risks allocating highly competitive places to students within these postal codes or schools who may come from better-off, educated families. The LMH FY team did considerable work, drawing on national policy, to develop a robust system for selecting low-SES students. Through literature review and an iterative process of testing, the LMH Foundation Year team identified a set of socio-economic indicators to act as proxies for educational disadvantage in the UK. Using these indicators enabled LMH to identify students whose socio-economic background was likely to have had an impact on their GCSE and A-Level grades. Current Higher Education policy suggests that such contextualisation is key to widening access [36,37]. See Table 4 below for more details.

While there are differences in the socio-economic assessment systems, there are some similarities. Both systems take a multi-indicator approach to assessing low SES. In both cases, this means that the targeted places within HE are more likely to be used for low-SES students. This individual targeting, or "contextualisation", was essential, as it enabled the LMH FY team to make a strong case to the university to modify the admissions requirements for the LMH FY, as it provided a holistic context for previous socio-economic circumstances having impacted on low-SES grade attainment. While there were concerns in the university that the assessment of both individual- and group-level proxies would be resource-intensive and unmanageable, it helped to be able to refer to HEAR in Ireland, where the application system is online, and documents are reviewed by a trained team nationwide. In Trinity, this multi-indicator assessment process began with the TAP Foundation Course in the early 2000s. The objective was to demonstrate to Trinity College Dublin that low-SES students could thrive in the university if given extra academic, personal and social support for a year prior to admission to first year, and that there was a robust alternative to identifying and supporting their potential. Within five years, the student academic attainment and retention within Trinity was so strong that this multi-indicator approach was tested for low-SES admission directly to the first year of university. By 2012, this developed into a national scheme, HEAR, alongside all other universities in Ireland. The collaboration between Trinity and LMH in Oxford enabled the use of this evidence base from Ireland to effect change in the University of Oxford. In some respects, LMH FY is now in the place TAP was in 20 years ago, aiming to scale the FY across the university and to establish a more robust approach to socio-economic assessment of low-SES students.

**Table 4.** The LMH FY Oxford University socio-economic indicators for low-SES students [36].

| Year 3: 2018/19 | | |
| :---: | :---: | :---: |
| **Compulsory Criteria** | | |
| **State-educated** | **Income Below GBP 42,875** | **NS-SEC 4–8** |
| **Indicators Used for Further Contextual Information:** | | |
| Receipt of free school meals | | |
| Parental education level | | |
| Index of multiple deprivation | | |
| Index of deprivation affecting children | | |
| POLAR quintile (school progression to HE) | | |
| ACORN category (post code progression to HE) | | |
| School admissions policy | | |
| % 5A*-A at GCSE school (attainment at age 16) | | |
| % 5A*-C at GCSE school (attainment at age 16) | | |
| Average point score at A-Level (attainment at age 18) | | |
| % of students in school receiving free school meals | | |
| School progression rates to Russell Group University | | |
| School progression rates to Oxford University | | |
| **OR** | | |
| Care leaver (at least six months in care of the local authority) | | |

A* denotes the highest academic grade a student may achieve in the English second level school system.

This section has described the development of socio-economic assessment systems within the context of admission to the University of Oxford and Trinity College Dublin. It explains how the evidence base from Trinity was used to strengthen the case for change within Oxford. It highlights how the indicators varied in both contexts, as they are drawn from nationally verifiable data sources; however, in both systems, a multi-indicator assessment approach is taken to better target the low-SES students who are most likely to need academic "contextualisation" and a year-long foundation course to address gaps in their schooling.

*4.2. Community-Focused Change, versus University-Focused Change*

The next point of comparison between Trinity and Oxford is in their approach to change. The TA21 programme emerged from the evidence base in Ireland regarding the grade attainment in higher education of low-SES students who had been given additional supports at entry to higher education. The students were found to perform as well, and in some cases better, than their non-low-SES counterparts [27–29]. This precipitated the TAP team to consider how much more untapped student potential was within their partner schools and to consider how to build a closer partnership with the schools to engage every student throughout their time in second-level schooling and to aim to address some of the gaps highlighted in the literature (limited educational guidance, an absence of educational role models, a limited range of subject choices, lack of money for additional educational support). This community-focused change has at its heart the concept of how best to use the university resources to support change within low-SES communities. The corollary of that is that students within those communities will be better prepared—in both information and academic attainment—to make informed decisions regarding their post-secondary trajectory, whatever that might be.

The LMH FY case is focused on institutional change within the University of Oxford. It aims for two substantial departures from practice: (a) to establish a university-wide foundation year, based on the success of the LMH FY and to admit low-SES students to

the course at lower academic attainment that those who progress directly to first year and (b) to change the socio-economic assessment process within the admissions process, to better target individual low-SES students. These are the changes that Trinity engaged in within the Irish context in the 2000s. Both are substantial changes to Oxford's admissions landscape, and they have already precipitated comparable changes within the University of Cambridge [38].

### 4.3. Evolution of Practice Based on Local Evidence Base versus Adaptation of Practice from Another Institution, Customized to Local Need

A third point of comparison is the use of evidence in both cases. TA21 combined the institutional evidence base for the success of low-SES students with the academic literature and policy indicators to determine how it could best have a "whole-school" effect within partner schools. This was an evolution of practice, which began with recruiting individual low-SES students for the TAP foundation course back in the 1990s. The greater their success, the more the team reflected that there must be many more talented, undersupported students in the schools, who could also benefit from progression to Trinity or another institution.

In LMH, there was a steep learning curve, as the leadership and college staff were committed to changing the socio-economic profile of the college and the university but did not have specific expertise in this field. The collaboration with Trinity enabled LMH to draw on its evidence base and to customize it for the Oxford context. This collaboration and "transfer" of expertise from Trinity to LMH over the first few years of the foundation year was pivotal in making the institutional change possible. Within Oxford, this meant the director of TAP and the LMH Foundation Year director addressing groups of fellows in different colleges within the university system, meeting with academics resistant to change on a one-to-one basis to discuss their concerns, addressing admissions committees, building external policy support for change through bodies such as The Sutton Trust and the OfS, and the principal of LMH leading the case for change at the heads-of-colleges level. The principal of LMH had himself been recruited from a major national newspaper to the position of principal, so he was acutely conscious of the rolling national debate regarding limited admission to the University of Oxford of low-SES students. The collaboration extended to building wider public support through media (radio, television, print media) and to bringing the TAP FC students and the LMH FY students together each year to consider how they could collaborate for change.

### 4.4. University as Change Agent versus University Responding to External Pressures

A final point of comparison is the position that each university took in their respective programs. In Trinity, the 20-year evidence base gave the institution confidence and experience to reach back into the second-level system and build a larger scale programme and to aim for system-level change—in creating strong whole-school, college-going cultures.

In LMH FY, there was some academic support for change but there was concern that it was not possible to recruit the "right" low-SES students and that the university would not be able to address gaps in schooling through a one-year course. There were also people within the university who believed that change was not necessary, and it was not the role of the university to "fix" gaps in schooling. There were external pressures on Oxford from the OfS, in respect of being able to charge full tuition fees, from the media, in highlighting continuing inequalities in its admissions, and from the "do-tank" The Sutton Trust, which had long called out the issues with Oxford and Cambridge admissions. The pressure for change within Oxford was mainly from external sources: policy, financing, media. The motivation for change within Trinity derived from the successful evidence base across the HE sector in preparing and admitting low-SES students, most of whom went on have high academic achievement in their degree courses. low SES.

This subsection has considered the first of two questions posed by the CCS, exploring changes to admissions processes in both institutions alongside a comparison of the two

programme approaches and the policy context within which they emerged. The following subsection seeks to answer the second research question.

*4.5. The Development of Capabilities in Students Engaged in Two University-Based Programs Aimed at Low-SES Students*

Until 2014, university-to-school educational outreach activities targeting low-SES students mainly adhered to a standard national model of providing some senior students from partner schools with opportunities to visit the university campus. Thanks to a significant external grant, this model changed in Trinity College Dublin, to focus on building the social, academic and human capital of *all students* within partner schools (grades 7–12) and providing professional development support for teachers to foster collaborative and reflective learning environments [28]. The first pilot phase of the initiative (Trinity Access 21 (TA21)) involved a three-year, quasi-experimental, intervention-style study, which ran until 2017 and followed a cohort of 1100 year 1 (grade 10) students from 11 treatment and 4 control schools (2 from areas of similar SES, as well as 2 with high progression rates to higher education). Results from this three-year pilot intervention indicate that the TA21 programme has a positive effect on participants' aspirations to continue in education after completing post-primary school, with evidence of increased aspirations and capabilities in the intervention group with respect to the control groups [25].

In addition to this overarching research study, Hannon undertook a qualitative longitudinal research (QLR) study on the program, which explored the question of *how social and academic supports for students, provided within the context of a school–university partnership, might contribute to the development of student capabilities.* The "capability approach" study used qualitative longitudinal research (QLR) a group of 35 research participants, who were students in the second year of secondary education at the outset (age 14). All students were from families where the parents did not initially progress to higher education, although two of the students had parents who had more recently undertaken higher education qualifications. The study began with students when they were 14 years of age and continued with the same students as they progressed through second-level education for the following 3 years (age 17). Student focus groups and interviews were triangulated with teacher focus groups and principal interviews. The study aimed to determine if there is a specific set of educational capabilities, the development of which would support *the capability to aspire towards post-secondary education* in low-SES students [24,25,33].

*4.6. Generating a Capabilities List*

Hannon [33] adapts a "top-down and bottom-up" approach to developing a capability list for low-SES student progression to post-secondary education from Wilson-Strydom's 2016 paper with a similar focus in South Africa [56]. Firstly, it proposes an ideal theoretical list [51], which combines the literature on access, widening participation and education policy with the body of research on using the capability approach in education (top-down).

- Stage one: developing an ideal theoretical capabilities list (top-down approach).

The development of a capabilities list for low-SES students' preparation for higher education progression began with a review of the literature on educational disadvantage, widening access and participation, social justice theories, theories of social and cultural reproduction and theories of critical pedagogy. These were considered in the context of the growing body of literature related to the capability approach and its usefulness in exploring education and, specifically, widening participation.

- Stage two: participatory-list development through qualitative longitudinal research.

This involved an empirical approach to the development of the list by interrogating its usefulness with young people in the four case-study schools. This engages with Sen's concern that it is critical to have a participatory process with those involved in the development of their own capability set. This research therefore involved young people in the discussion and development of the capability list for higher education progression through

a qualitative-longitudinal research approach and specifically focused on their engagement with the TA21 programme. Student interviews and focus groups were designed with the ideal theoretical capability list in mind, and they involved discussions with the students of "valued doings and beings", to explore the development of capabilities relevant to education and their ability to convert these to valued functionings. The capabilities that emerged frequently in thematic analysis are not all-encompassing of the young people's lives; they are focused on the impact of the TA21 programme, as that is the focus of the evaluative lens of this study.

Having adapted Wilson-Strydom's [56] "top down/bottom up" approach, it is proposed that there are five key capabilities which students develop through their engagement in this university-to-school partnership programme. These capabilities are enriching student ability to make informed choices about their future, to feel more autonomous as young adults, to build trusted networks of relationships across their communities and to engage constructively with their own "identities in flux" to refine and embellish their hopes for the future. The capabilities emerging through this application of the capability approach are defined in Table 5 below:

**Table 5.** A pragmatic capabilities list to prepare low-SES students to aspire towards higher education [25,33].

| Capability | Definition |
|---|---|
| 1. Autonomy | Being able to have choices, having information on which to make choices, planning a life after school, reflection, independence, empowerment. |
| 2. Practical reason and college knowledge | Being able to make well-reasoned, informed, critical, independent and reflective choices about post-school study and career options. Knowledge is system knowledge, rather than academic skills or abilities, and is aligned to practical reason such as the capability to assess and evaluate this new knowledge base and incorporate it into a new frame of reference. |
| 3. Identity | Identity as a matter of "becoming" as well as "being", belonging to the future and the past, taking place in the spaces of relations in which individuals are embedded. |
| 4. Social relations and social networks | The capability to work with others to solve problems or tasks. Being able to form networks of friendship, belonging and mutual trust to support the development of navigational capital for progression to higher education. |
| 5. Hope | Aspiration, motivation to learn and succeed, to have a better life, to hope. |

- Conversion Factors

The pragmatic capability list provides a framework for what is required to successfully enable low-SES students to aspire to higher education; however, as Wilson-Strydom [56] observes, it is also important to identify conversion factors that impact a person's ability to convert resources into opportunities or capabilities. This draws attention to the point at which agency and structure intersect and therefore provides a mechanism to explore how individual agents can engage with positive or negative structural processes, or conversion factors, to realise their goals. Table 6 below provides an overview of the three TA21 core practices and the capabilities that emerged over the three-year period (2014–2017).

**Table 6.** The TA21 core practices and emergent student capabilities [33].

| Valued Capabilities Identified through the Research and Related Social and Cultural Capital Themes | TA21 Core Practice | Example of Activity |
|---|---|---|
| **Autonomy, practical reason and knowledge, identity, hope** Understanding college costs Understanding career pathways | Pathways to College | College visits, gaining course knowledge, building information on financial processes. |
| **Autonomy, practical reason and knowledge, social relations and social networks** College and career knowledge Overcoming barriers Goal-setting | Leadership | Redeveloping an unused school room as a 21st century learning space, identifying funding sources, project planning and implementation. |
| **Social relations and social networks, hope, identity** Networks Trustworthy information Navigational capital/educational resilience | Mentoring | Total of 6 structured mentoring workshops per year with a mentor who has recently progressed from local school to college. |

The research demonstrated three main findings: (1) Specific student capabilities emerged following their engagement in the core practices of Leadership, Mentoring and Pathways to college; these are: autonomy, practical reason and college knowledge, identity, social relations and networks and hope. (2) Students encountered a range of inhibiting social conversion factors in developing capabilities and persisting with higher-education aspirations; these were: the negative pull of peer relations, pressure related to state examinations, and limited subject choice and conflicting family expectations. (3) There are four themes that arise throughout: (a) the centrality of informed choice, (b) confidence, (c) resilience and (d) trusted relationships with relatable others.

Having access to early information about subject choice and levels enabled them to consider their educational choices in the long term and understand how thinking about their future now might positively impact their progress. The Leadership core practice helped students to develop the confidence and ability to "ask the right questions". This also enabled them to broaden their networks outside of the immediate community, through campus visits and business presentations. Students built new, trusted relationships with their mentors, which made them believe that *"if they can do it, then so can I!"* This encouraged them to discuss their future with their families and teachers, which in turn, improved the quality of those relationships. Having access to relatable others and to information on the variety of entry routes to post-secondary education supported students' educational resilience, as they internalised the message that there are many ways in which they could progress and that they could draw on their social networks [33].

*4.7. The Development of Capabilities in Students Engaged in the LMH FY*

O'Sullivan et al. [34,35], a research team from Maynooth University and the University of Oxford, undertook a longitudinal comparative case study of the student experience on the LMH FY. Pre- and post-questionnaires and in-depth focus groups were undertaken with the FY students. Analysis of the focus groups conducted at the start of the academic year explored students' experiences before entering university and perceptions of why they needed an alternative entry route to the university. Ten students from the LMH FY participated. They completed a questionnaire at the start and the end of the year to determine their perceived levels of academic capital and sense of belonging in the university. Qualitative data were analysed using interpretative phenomenological analysis [34], a qualitative approach which explores in detail how participants are making sense of their personal and social world.

Similar to the second-level students in the TA21 programme, O'Sullivan [34,35] reports strong feelings of inequality in student views of the education system. They perceived some students as having a natural sense of belonging, while others believed the lack of educational guidance and knowledge in their school meant they had a lack of "navigational capital" to support their progression to HE. Another similarity in the findings both in Ireland and the UK is that students in the LMH FY referenced the importance of having relatable role models within their family and community to build this sense of confidence, belonging and identity as a HE student.

Thematic analysis of the focus groups conducted at the end of the LMH FY explored students' experiences of the LMH FY, their perceptions of how it impacted upon their capacity to participate in university life and any challenges students faced over the year. The analysis identified the processes of change which occurred over the course of the year and four themes that arose, related to student identity, belonging and confidence [34,35].

This article draws on the findings of the O'Sullivan et al. paper as a basis for comparison with the TA21 programme to consider the capabilities that emerged in students during their participation in both programs. In Table 7 below, a capability lens is applied to the qualitative findings of O'Sullivan et al.'s research and the emergent capabilities of academic identity, affiliation and practical reason that are aligned with the LMH component and practical examples.

**Table 7.** The LMH FY in Oxford University and emergent student capabilities.

| Valued Capabilities Identified through the Research and Related Social and Cultural Capital Themes | LMH FY Component | Example of Activity |
|---|---|---|
| **Identity, Social Relations and Social Networks, Practical Reason.** Understanding academic processes. Managing college finances. Building social networks. Knowledge development and cultural engagement. Sense of belonging and confidence. | Tutoring system. Preparation for undergraduate study course. Course participation. Extra-curricular cultural and social events. Academic assessment and feedback. | Attending course on study skills. Information sessions on academic processes. Attending university events and talks. Participating in student-led events and student council. |

*Identity*. FY students described a significant shift in how they viewed their academic potential. Academic courses, focused on essay writing and critical skills, alongside a supportive, accessible tutoring system, changed students' perception of their academic identity. This helped students to grow in self-confidence. The process of drafting academic essays, presenting and defending their academic ideas with others, further strengthened this emergent identity as a member of the LMH college community and as students within the University of Oxford. The tutorial system in Oxford, which gives each student a close academic advisor and support, was essential in developing the students' identity, their academic confidence and their sense of belonging.

*Practical Reason and College Knowledge.* The knowledge developed through small-cohort class discussions with tutors and professional staff within the college helped students to build the capability of practical reason to plan their future and to assess their own strengths and abilities within the University of Oxford context.

*Social Relations and Social networks*. FY students formed a strong community amongst themselves; however, they were also integrated within LMH as part of the college community, and within the University of Oxford as part of the wider institutional community. As the first FY cohort in the university, they were subject to considerable scrutiny as an "admissions experiment". There were sceptics both among the staff and student bodies. In-depth work between the LMH team and the FY students was essential to teasing these complexities out, enabling them to explore and discuss their social class identities, inequali-

ties within the wider system and to build an understanding of their legitimate place within the university. This process strengthened FY students' capability of social relations and social networks, by building a stronger sense of collaboration with their peers and their tutors, enabling them to reach beyond their own networks into the wider university community, and exposing them to public discussions, fora and cultural events which enhanced their sense of belonging. The LMH FY students were all high academic achievers before they began the course; however, O'Sullivan et al. [34] demonstrated that this academic success masked a sense of "outsider" within the university and the process of engaging in the FY helped students to develop the capabilities of identity and social networks within the university.

## 5. Conclusions

This article provides an overview of how countries with common challenges regarding the participation rate of low-SES students in higher education have partnered in practical and academic ways to adapt and scale effective models and to build the evidence base for further learning. It has used the capability approach as a lens through which to consider the impact of the programs on participating students. These programs and associated research would not have happened without an openness by the institutions, the partner schools and the students to learn from each other and collaborate to address the common challenge of unequal access to higher education. This comparative case study (CCS) poses two questions:

(1) How did the admissions processes in both universities change to target low-SES students?
(2) How do social and academic support services for low-SES students, provided by two universities, contribute to the development of student capabilities?

### 5.1. Admissions Changes

The CCS have undertaken, in the context of a collaboration between Trinity College Dublin and LMH Oxford University (2016–2020), the development of a foundation year and an adapted admissions process.

Admissions processes in both institutions changed to target low-SES students. The evidence of strong academic performance by such students in the Irish context helped to strengthen the case for more targeted admissions at lower academic thresholds in the UK context. Both institutions drew on nationally available proxies for educational disadvantage. In Ireland, students admitted to Trinity and the other Irish universities now have a long and strong track record of academic progression and achievement in higher education, and they continue to make the case for diversified admissions, that takes account of the fact that student potential may be masked or underestimated due to socio-economic circumstances.

By 2022, 43 former LMH FY students had matriculated at the University of Oxford. Thirteen former Foundation Year students had graduated with degrees from Oxford University including two with First Class Honours. The student success on the LMH FY echoes the strong academic attainment and higher education progression of the TAP FC and HEAR students in Ireland [36].

The persistence and success of the LMH FY students, along with external policy pressure from the OfS, helped to make the case for Oxford University to scale the FY project beyond LMH. Between 2019 and 2020, both Oxford and Cambridge Universities announced their intentions to develop a university-wide FY to diversify their socio-economic intake, with an objective that one in four of their admissions would be from low-SES groups by 2025 [36]. The University of Cambridge admitted 43 low-SES students to its first Foundation Year in 2022, and the University of Oxford will admit 50 students to an institution-wide programme in September 2023 [38,39]. This is the biggest shift in the Oxbridge admissions landscape since the awarding of degrees to women over a hundred years ago.

### 5.2. Community-Based Change

The TA21 programme emerged from 20 years of practice, which demonstrated that low-SES students could survive and thrive in selective HEIs. By 2013, the TAP team considered what more they could do to develop the potential of all students within partner schools and to tackle some of the persistent challenges identified in the literature. TA21 aimed to build a whole-school college-going culture and to engage as many students, staff and community networks in this objective as possible. In 2014, 48% of students from TA21-linked schools progressed to higher education. Over time, this number has gradually increased and in 2020, the schools with high levels of engagement are reporting up to 74% of their students progressing to HE [30].

The LMH and Oxford approach focused much more on bringing 16–18-year-olds to the campus for summer schools and other outreach programs, but it was not aiming for this deeper level of community-based change. It took an individual approach to selecting low-SES students rather than a broader structural approach that considered the students' lived realities. In this respect, the development of thinking and associated practices within Oxford are at a much earlier stage of development than in Trinity.

As the former Principal of LMH, Alan Rusbridger observed, "If Oxford shrugs—"we can't find them either!"—and blames a failing school system, it will look to some as if it's failing in its wider social, educational and charitable purposes" [40]. The Oxford response was, in many respects, a consequence of public policy pressures and, within LMH, emerged from a genuine desire to create greater socio-economic diversity and respond to its historic mission as the first women's college. The Trinity response, through TA21, evolved from institutional learning about the potential and success of thousands of students from low-SES backgrounds and a commitment to aim for larger-scale community-based change. These differences provide insight into diversification strategies in two of the world's most selective universities.

### 5.3. A Capability Approach to Exploring Change

Using the capability approach to explore outcomes on these two programs enables us to visualise the impact widening participation interventions might have if they were designed to widen the capability of low-SES students towards valued beings and doings rather than a principal focus on ensuring more students' complete schooling with the attainment required to progress to higher education. Its conceptual contribution is therefore to combine the capability approach with theories of social and cultural reproduction, social justice and critical pedagogy to develop a pragmatic list to best prepare low-SES students to aspire towards higher education. It enables us to build a picture over time of what student capabilities are emerging and how these may be either strengthened or diminished by social, personal or environmental conversion factors.

The "capability approach" framing contends that if educational interventions had as their starting point the objective of empowering students to develop to their full potential, through a focus on the capabilities they need to continue to aspire, then students would be more likely to develop educational resilience and navigational capital to persist and to focus on their longer-term goals, even in the context of environmental adversity. It also proposes that students who have the knowledge they need to make informed choices, particularly at an earlier stage in their second level education, are less likely to be inhibited in their decision making by adapted preferences and, whether they choose to progress to higher education, are at least making their choices in the context of a more complete information base and relatable, college-going networks.

In both programs, this individualised approach to exploring student capability development, and to socio-economic assessment, factors in that each individual is unique within the blunt scope of policy formation and that programs and evaluations can be designed to take into account this potential and the possibilities of change through collaboration.

This article began by providing the context for widening participation in Ireland and the UK. It proceeded with an explanation of two programs and the admissions changes

that took place as a result of collaboration and shared learning. It provided an overview of qualitative research that informed this comparative case study and applied a "capability lens" to exploring the data.

Drawing on a community-based programme in Ireland, the article explained the impact of a whole-school partnership approach to low=SES student capability development and contrasted this with a university-focused academic programme, aiming to diversify student intake in Oxford University. The Trinity strategy had moved beyond a consideration of its own student profile to the potential impact the university could have within communities. The Oxford strategy resembled the Trinity strategy in the early 2000s in that its focus is on individual student recruitment from low-SES backgrounds and efforts to reform the institutional admissions system. Both programs are examples of the power of learning across geographical contexts and collaborative action. The two programs share a focus on building human capability as a way of improving the realisation of opportunities for the many capable, committed and challenged low-SES young people, who are often overlooked and underestimated by the mainstream-schooling and higher-education selection systems.

**Funding:** This research received no external funding.

**Institutional Review Board Statement:** Ethical approval for this study was gained from the Faculty of Arts, Humanities and Social Science Ethics Committee in Trinity College Dublin, the University of Dublin on 20th March 2015.

**Informed Consent Statement:** Informed consent was obtained from all subjects involved in the study.

**Data Availability Statement:** Data is unavailable due to ethical considerations, as the research involved children.

**Conflicts of Interest:** The author declares no conflict of interest.

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
