# Peer review of "A Human Capability Perspective on the Progression of Low-SES Students to Higher Education in Ireland and the UK"

_education, doi:10.3390/educsci13040409_

Round 1

Reviewer 1 Report

Article Summary

The article focuses on social action to target and develop human capabilities in low ses communities. The authors present an intervention designed to improve the understanding and realization of opportunities by those marginalized in the educational system. They detail the use of this approach to compare and contrast a whole school partnership and a university focused program. The authors present an exciting and important idea in the context of supporting educational equity however as written the is hard to follow.

Major Points:

To help the readers the authors should more strongly motivate the study involved in widening capability and the important research questions in the introduction. The intro didn’t seem to strongly motivate the approach and method clearly and succinctly.

It was hard to understand the research questions and methods being applied here because there was a great deal of coverage of large literature, but this coverage did not correspond with clear flow from one section to the next.  As a reader I was often unclear on why I was reading these sections. The state of the current draft doesn’t clearly motivate the current research in the context of developing an argument for the work being researched.

Likewise, the purpose of the research and findings should be streamlined and articulated better in the discussion. The discussion should have a thorough coverage of what was main question how it was tested and what was found. Also, suggest including how this project expands research and theory and what future directions should be.

Minor Points

Why are there tables and such in the discussion? Maybe it’s a style difference but I expected all the results main findings, and any visuals should be included in the methods or results sections.

Recommendation

This article just did not come together to me as a coherent whole. I’d advise clarifying the major and minor points included to strengthen the manuscript. I was left unclear about what could be learned about this approach after reading the article and I was unsure on exactly how to place this work in a larger literature. It was difficult to understand what specifically was learned and what the next step in future research in this area might be. Considering these major and minor points I’d suggest that this article needs a major revision. Only after a major revision could the article be reconsidered to determine whether it might be acceptable for publication.

Author Response

I attach a note to Reviewer 1.

Reviewer 2 Report

Thank you for the possibility to review this paper.

The topic is interesting and relevant.

As a reviewer, I would suggest the following:

- to slightly rearrange the Abstract and Introduction parts.

- to improve the introduction by adding traditionally included parts, such as the aim of the study, research questions, and summary of the following sections.

- to elaborate on the recommendations that can be taken away from the study.

Author Response

I attach responses to Reviewer 2.

Reviewer 3 Report

Overall, it is a good study. There is a need to clearly define the research methodology for this comparative study and it's linked with socio-economic status. You need to clearly identify that what is the impact of low SES on Higher education and the student performance. Though you have mentioned the Human Capital and Human Capability approach, but it's linked with both sample schools is missed. The quality of this article can be improved if you can add some information about Oxford strategy and Trinity strategy as i think it will be helpful for the reader. while discussing the Comparing change between a community-based and a university-based education intervention, I think you can add some more details about the community as background information and salient features (table -3)  

Author Response

I attach responses to Reviewer 3.

Reviewer 4 Report

First of all, I would like to thank the authors for the opportunity to read this paper. The subject is both interesting and engaging, still, there are a few aspects that the authors should address, in order to enhance the quality of the article.

The paper entitled "A Human Capability Perspective on the Progression of Low 2 SES Students to Higher Education in Ireland and the UK" presents relevant information for the field. The abstract should also briefly deliver a few results obtained after the study.

The  Introduction and theoretical background present sufficient information on the subject, but I didn't identify the hypothesis and the objectives of the study. Are they supported by the results obtained?

The conclusions section could be extended.

The references should be update, as 187 sources from 208 are older than 2018...

Author Response

I attach responses to Reviewer 4.

Round 2

Reviewer 4 Report

The authors addressed all the points in the first review and made the necessary changes.
Best regards!